# Numerical Simulation of Integrated Generation and Shaping of Airy and Bessel Vortex Beams Based on All-Dielectric Metasurface

**DOI:** 10.3390/nano13061094

**Published:** 2023-03-17

**Authors:** Kuangling Guo, Yue Liu, Zhongchao Wei, Hongzhan Liu

**Affiliations:** Guangdong Provincial Key Laboratory of Nanophotonic Functional Materials and Devices, School for Information and Optoelectronic Science and Engineering, South China Normal University, 378 Waihuan West Road Panyu District, Guangzhou 510006, China

**Keywords:** multifunctional metasurfaces, optical vortex, Airy beams, cross-phase

## Abstract

Integrating multiple independent functions into a single optical component is one of the most important topics in research on photoelectric systems. In this paper, we propose a multifunctional all-dielectric metasurface that can achieve a variety of non-diffractive beams depending on the polarization state of the incident light. Using the anisotropic TiO_2_ rectangular column as the unit structure, the three functions of generating polygonal Bessel vortex beams under left-handed circularly polarized incidence, Airy vortex beams under right-handed circularly polarized incidence and polygonal Airy vortex-like beams under linearly polarized incidence are realized. In addition, the number of polygonal beam sides and the position of focal plane can be adjusted. The device could facilitate further developments in scaling complex integrated optical systems and fabricating efficient multifunctional components.

## 1. Introduction

In 1992, Allen et al. [1] discovered for the first time that photons can carry orbital angular momentum (OAM) in addition to general spin angular momentum (SAM) [2], which laid the foundation for the study of vortex beams. Optical vortex beam (OVB) is a special beam carrying an OAM of *lħ* per photon. Due to its singularity effect, OVB has a helical phase exp(i*lφ*) in the wavefront structure, where *l* is the topological charge (TC) number which can take any integer value, *φ* denotes the azimuthal angle and *ħ* is the Planck constant. Since then, OVB has been widely used in quantum information processing [3,4], quantum entanglement [5], optical micromanipulation [6,7], optical communication [8,9,10] and nonlinear optics [11,12]. Hence, with the increasing demand for practical applications, OVB needs to be further studied: from generating [13,14,15] and measuring [16] to shaping OVB [17]. Especially in recent years, shaping OVB has a research hot spot. Current common shaping methods such as multi-OVB interaction [18,19,20] and optical oscillatory elements [21] require complex holography or high-precision diffractive elements. Until 2019, a new type of phase structure called higher-order cross-phase (HOCP) was introduced into the modulation of the beam. The function of HOCP is to introduce astigmatism into OVB in phase, so that the optical field mode changes during the propagation process, thereby changing the shape of OVB [22]. Compared with previous methods, it has higher integration and simpler implementation, which opens up a new horizon for shaping OVB.

However, in practical applications, a single OVB will be affected by many clutters in the propagation, which will deteriorate the performance. In order to further stabilize and improve the performance of OVB, a combination of OVB with other beams is a solution. In 1979, Berry and Balazs [23] first theoretically proved that the Schrödinger equation has a solution in the form of an Airy function and predicted a non-diffractive Airy wave packet. The optical version of the Airy wave packet is known as the Airy beam. It has the same properties of propagation intensity distribution invariance [24], non-diffraction [25,26] and self-healing [27] as the traditional non-diffracting beams (such as Bessel beam). However, unlike traditional non-diffracting beams, Airy beam can undergo a lateral self-accelerating process without any external force [28], which is one of its most remarkable features. Although Airy beam possesses novel propagation properties, theoretically, infinite energy is required to produce Airy beam. Therefore, it was not until 2007, after the first experimental demonstration that Airy beam can be realized in finite energy by Civiroglu et al. [29], that Airy beam started to be used in many studies, including the Airy laser [30,31,32], particle manipulation [33,34,35], space-time wave packets [36,37] and so on. Further, by superimposing the vortex phase on Airy beam, the Airy vortex beam can be generated. As a special vortex, Airy vortex beam combines the dual characteristics of Airy beam and vortex singularity, and it has good prospects in medical treatment [38]. Traditional methods of generating Airy beams and vortex beams rely on the use of spatial light modulators [39,40] or diffractive optical elements [41,42], among others. Nonetheless, the above methods are often greatly limited in practical applications due to their complex structures or low efficiency.

As a reduced-dimensional representation of metamaterials, the two-dimensional (2D) metasurface is composed of subwavelength structures that can flexibly manipulate the amplitude, phase and polarization of incident light waves, and it has developed rapidly in recent years. Compared with traditional bulk optical elements, metasurface has the advantages of small size and flexible design adjustment. Due to its unique physical properties, metasurface has been widely used in various applications such as anomalous refraction and reflection [43], spin Hall effect [44,45], stealth [46] and hologram [47,48]. Since optical metasurface is sensitive to the polarization state of light, a series of polarization-sensitive multifunctional metasurface has been extensively explored and studied by taking advantage of this feature. For example, Wen et al. realized a multifunctional metasurface which could produce a single Airy beam and arrays of Airy beams in 2020 [49]. Li et al. formed a vortex beam with multi-channel polarization conversion on a single metasurface in 2021 [50]. These two studies not only prove the enthusiasm of research on multi-channel metasurface, but also they demonstrate the needs of generating Airy beams and vortex beams in light field regulation.

Combining with HOCP, this paper proposes an all-dielectric multifunctional metasurface to realize multiple polygonal vortex beams, as shown in Figure 1. The design of the metasurface is based on the use of anisotropic TiO_2_ structure as a unit. To achieve an adjustable polarization sensitivity difference in the x and y directions, the phase of transmitted light is modulated by the combination of the PB phase and the propagation phase. The polarization principle of the PB phase can be understood by the Poincaré sphere. Using the conventional visible light wavelength *(λ* = 532.8 nm) as incident wavelength, under the incidence of right-handed circularly polarized (RCP) light, the generation of polygonal Bessel beam can be realized based on HOCP, where the number of the sides of polygonal beam is equal to the order of HOCP. Additionally, a non-diffracting Airy beam with a controllable focus position is generated in the case of the left-handed circularly polarized (LCP) light incidence. Moreover, under the linearly polarized (LP) incidence, the amplitudes and phases of the two beams are superimposed on each other, so both the Airy vortex-like beam and polygonal shaping can be realized. We only need to change the order of the HOCP to obtain a desired polygonal vortex beam, which greatly simplifies the shaping process. Compared with other multifunctional metasurfaces, we only use the difference in polarization sensitivity to realize the generation of three kinds of non-diffracting beams, and add beam shaping function on the basis of traditional vortex, which has certain reference significance for the development of multifunctional metasurfaces. Further, since the modulated beam has a stable distribution at the far-field focus, it is of great significance to particle trapping, 3D optical tweezers, parallel laser printing and multiplex fluorescence imaging.

## 2. Theoretical Analysis and Design Method

### 2.1. Design of Multifunctional Metasurface

In this section, we will describe the physical mechanism that achieves the polarization sensitivity difference in the *x* and *y* direction in the transmission mode. We consider an arbitrary anisotropic nanopillar unit placed parallel to the *z*-axis, with the long and short axes along the *x* and *y* directions, respectively. When the unit is excited by the incident light with different polarization states, it produces responses with different phases and amplitudes along the *x* and *y* directions. Under LP light incidence, the Jones matrix of the transmitted field for an arbitrarily anisotropic unit rotated by an angle around the z-axis in the xoy plane is expressed as follows:(1)$$\begin{array}{l} T(x,y)=R(-\theta )\left(\begin{array}{cc} T_{x}e^{i\varphi_{x}} & 0\\ 0 & T_{y}e^{i\varphi_{y}} \end{array}\right)R(\theta )\\ =\left(\begin{array}{cc} \cos\theta & -\sin\theta \\ \sin\theta & \cos\theta \end{array}\right)\left(\begin{array}{cc} T_{x}e^{i\varphi_{x}} & 0\\ 0 & T_{y}e^{i\varphi_{y}} \end{array}\right)\left(\begin{array}{cc} \cos\theta & \sin\theta \\ -\sin\theta & \cos\theta \end{array}\right) \end{array}$$
where *T_x_*, *T_y_*, *φ_x_* and *φ_y_* represent the transmission amplitudes and phases along the *x* and *y* axes. *θ* is the rotation angle of the unit relative to the reference coordinate system. *R* represents a 2 × 2 rotation matrix. Here, we assume that the transmission amplitude of the unit remains unity amplitude (*T_x_* = *T_y_* = 1) with a π phase difference (*φ_x_* = *φ_y_* + π) for the *x* and *y* polarization cases.

The independent phase modulation of LCP and RCP on the metasurface can realize the generation of *φ*_1_(*x*, *y*) and *φ*_2_(*x*, *y*) two different phase distribution beams. This means that when the RCP is incident, the metasurface will only be modulated by *φ*_1_(*x*, *y*). Similarly, the metasurface will only generate a beam modulated by *φ*_2_(*x*, *y*) when the LCP is incident. This modulation mechanism on metasurface can be represented by a matrix *J*(*x*, *y*):(2)$$J(x,y)|LCP\rangle =\exp[i\varphi_{1}(x,y)]|RCP\rangle$$
(3)$$J(x,y)|RCP\rangle =\exp[i\varphi_{2}(x,y)]|LCP\rangle$$

The matrix form of |LCP〉 corresponds to $\left[\begin{array}{c} 1\\ i \end{array}\right]$, and |RCP〉 corresponds to $\left[\begin{array}{c} 1\\ -i \end{array}\right]$. After the matrix transformation in Equations (2) and (3), the matrix *J*(*x*, *y*) can be expressed as:(4)$$J(x,y)=\left[\begin{array}{cc} e^{i\varphi_{1}(x,y)} & e^{i\varphi_{2}(x,y)}\\ -ie^{i\varphi_{1}(x,y)} & ie^{i\varphi_{2}(x,y)} \end{array}\right]{\left[\begin{array}{cc} 1 & 1\\ i & -i \end{array}\right]}^{-1}$$

Then, by making the transmission matrix *T*(*x*, *y*) of Equation (1) equivalent to the modulation matrix *J*(*x*, *y*) of Equation (4), the phase distribution along the *x* and *y* directions and the unit rotation angle can be calculated, respectively.
(5)$$\varphi_{x}(x,y)=[\varphi_{1}(x,y)+\varphi_{2}(x,y)]/2$$
(6)$$\varphi_{y}(x,y)=[\varphi_{1}(x,y)+\varphi_{2}(x,y)]/2-\pi$$
(7)$$\theta (x,y)=[\varphi_{1}(x,y)-\varphi_{2}(x,y)]/4$$

The arrangement of units in the metasurface is determined by Equations (5)–(7). The phase design is mainly determined by the combination of the Pancharatnam–Berry (PB) phase adjustment based on Equation (5) and the propagation phase adjustment based on Equation (7). The PB phase is based on the adjustment of the phase by rotating the direction of the half-wave plate under circularly polarized light, which can be explained by the Poincaré sphere [51]. As shown in Figure 2, the north and south levels of the Poincaré sphere, respectively, represent LCP and RCP, and each point on the equator line corresponds to a different linear polarization state, while the entire spherical surface The other points represent elliptical polarization states for different polarization states. When a point on the sphere rotates an angle *Ψ* with the center of the sphere as the center, the phase change shown on the sphere is twice the rotation angle, which is the three-dimensional interpretation of the geometric phase. Therefore, when the rotation angle of nanopillars varies from 0 to π, the PB phase of 0 to 2π can be obtained. Since this phase is only related to the orientation and size of units, and since the different polarization states of the incident light will produce different phase responses, this feature can be used to perform differential encoding of left-handed and right-handed incidence, thereby realizing multifunctional metasurfaces.

### 2.2. Design of the Unit Structure

In order to better realize the combination of the transmission phase and the PB phase, the design of the units in the metasurface needs to be adjusted by the rotation angle and size. The design of a unit is shown in Figure 3. Due to the high refractive index and relatively low loss of TiO_2_ dielectric material at visible wavelength, TiO_2_ semiconductor is selected as the rectangular unit material which placed on the SiO_2_ substrate. The refractive index parameters (*n*, *k*) of both TiO_2_ and SiO_2_ materials are taken from [52]. To further discover the realization mechanism of phase, we performed numerical simulations using a three dimensional finite difference time domain (FDTD) solutions. We use scripts to import the above theoretical content into FDTD, and obtain corresponding results under 3D simulation to support the above theoretical derivation. The size of a mesh cell we set is 0.02 µm, and the simulation wavelength is set as *λ* = 532 nm. In the case where the simulation sets this value to the wavelength condition, the real part (*n*) of refractive indices of TiO_2_ and SiO_2_ are $n_{TiO_{2}}=2.17$ and $n_{SiO_{2}}=1.48$, and the imaginary part (*k*) of the refractive indexes are both approximately zero. Considering both integration and transmission effects in this design, we set H = 900 nm and P = 400 nm, and *L* and *W* were varied from 50 nm to 400 nm, respectively. Figure 3b shows the normalized transmittance and phase magnitude of the 11 units selected after the simulation. It can be clearly seen that the selected units satisfy the full coverage of the 2π range well, and has a constant phase response difference of approximately π for *x* and *y* polarization. We apply periodic boundary conditions in the *x* and *y* directions and implement a perfectly matched layer (PML) in the *z* direction. The unit can be thought as a rectangular waveguide with birefringent properties due to the asymmetry of its geometry. Therefore, the phase delay along the *x* and *y* axes can be achieved by changing *L* and *W*.

The 11 black dots in Figure 4 correspond to the units in Figure 3b from left to right and from top to bottom. Figure 4a,c depict the relationship between the size (*L*, *W*) of units and the phase shift and transmission coefficient under *x*-LP light, respectively. It can also be seen that the phase retardation can well span the entire 2π range, and the transmittance is always at a high value of more than 80%. Therefore, the desired phase combination can be obtained by rationally choosing the unit size. In addition, the selected units are well satisfied that the phase difference are approximately π under *x*-LP and *y*-LP incidence, which can be clearly drawn from Figure 4b. The calculated phase shifts and transmission coefficients of individual TiO_2_ unit under *x*-LP and *y*-LP light incidence are presented in Figure 4d. In a word, the independent regulation of RCP and LCP can be well achieved by all units.

## 3. Results and Discussion

### 3.1. Polygonal Bessel Vortex Beams

The phase *φ*_1_(*x*, *y*) of the cross phase (CP) in Cartesian coordinates (*x*, *y*) has the form [22]:(8)$$\varphi_{1}(x,y)=u\cdot (x^{p}\cdot \cos\theta -y^{q}\cdot \sin\theta )\cdot (x^{p}\cdot \sin\theta +y^{q}\cdot \cos\theta )$$
where the parameter *u* controls the conversion rate, the azimuth factor *θ* characterizes the rotation angle of the transformed beam on the Fourier plane, *p* and *q* are arbitrary positive integers, and the value of *p* plus *q* is equal to the order of CP. When CP is low-order cross phase (LOCP), that is, both *p* and *q* have a value of 1, the number of topological charges can be determined. Further, when the sum of *p* and *q* exceeds 2, CP as the high-order cross phase (HOCP) can realize adjusting the beam to an arbitrary polygon. This paper mainly exploits the ability of the CP to shape the beams, so the HOCP mode is adopted.

For computational convenience, we consider the case where the CP does not rotate (*θ* = 0). The phase *φ_x_*(*x*, *y*) of adding HOCP to the Bessel vortex beam is:(9)$$\varphi_{x}(x,y)=-\frac{2\pi }{\lambda }\cdot \sqrt{x^{2}+y^{2}}\sin(\beta )+l\cdot \arctan(\frac{y}{x})+u\cdot x^{p}\cdot y^{q}+2\cdot \eta \cdot \pi$$
where *λ* represents the incident wavelength, *β* represents the angle between the transmitted wave and *z*-axis (set to 12.7°), *l* represents the topological charge, the parameter *u* controls the beam conversion rate and *η* is an arbitrary real number. It can be seen from Equation (9) that the magnitude of the value of *u* is affected by the order of CP. Since the designed structure is nanoscale, the value of *u* will be relatively large. As an example, in the following Figure 5, the generation process of the phase mask composed of the HOCP and the 2nd-order Bessel vortex beam is shown. Figure 5a_1_–a_4_ corresponds to the phase masks from the 3rd-order CP to the 6th-order CP, respectively, and Figure 5b_1_–b_4_ shows the phase of the 2nd-order Bessel vortex beam after superimposing HOCP in Figure 5a_1_–a_4_. It can be clearly seen that the Bessel vortex beam can be shaped into any polygon after adding HOCP, where the number of polygon sides is equal to the order of HOCP. Another way of saying this is that when the order of HOCP is three, four, five and six, the resulting OVB is shaped into triangular, quadrilateral, pentagonal and hexagonal polygons.

Based on the above theory, considering the influence of *p* and *q* values on beam shaping, the *xoy* plane electric field simulated at the focal position under RCP incidence is shown in Figure 6. Figure 6a_1_–d_1_ represents the electric field and phase distribution of the 2nd-order and 4th-order Bessel vortex beams without adding HOCP, respectively. It can be observed that Bessel vortex beams of any order can be well generated. When HOCP is added, the electric field presents a polygonal ring of corresponding order, and the phase presents a polygonal distribution. Among them, some basic properties are the same as the traditional Bessel vortex light, for example: the radius of the beam will increase with the increase in the order of the Bessel vortex beam, and there will be secondary halos of various levels outside the primary halo. From Figure 6b_1_–b_5_ and Figure 6d_1_–d_5_, it can be clearly known that the adjustment of HOCP is based on the phase. By polygonal shaping the vortex part of the phase mask, the shape of the spatial beam can be changed, which can confirm the numerical results derived in Figure 5. Since the manipulated particle will move in the direction of the energy flow, as long as the values of *p* and *q* are controlled according to the desired result, the motion trajectory of the manipulated instance can be accurately determined.

### 3.2. Airy Beams

Secondly, the phase of the Airy beam is:(10)$$\varphi_{y}(x,y)=\frac{\tau^{3}}{3}(m^{3}x^{3}+n^{3}y^{3})-\pi$$
where *τ* = *k*/*f*, *k* = 2π/*λ*, *λ* represents the wavelength of the incident light, *f* is the focused wavelength, and *m* and *n* are arbitrary constants. To verify that the phase structure can realized the Airy beams, we assume the complex amplitude of the plane wave moving along the *z*-axis:(11)$$\vec{E}(x_{0},y_{0},z)=A\exp(ikz)$$

After Fraunhofer diffraction of light field, the expression for the outgoing electric field is:(12)$$\vec{E}(x,y)=\frac{\exp(ikz_{1})}{i\lambda z_{1}}\cdot \exp[\frac{ik}{2z_{1}}(x^{2}+y^{2})]\cdot \iint_{\sum }\vec{E}(x_{0},y_{0},z)\cdot \exp[-\frac{ik}{z_{1}}(xx_{1}+yy_{1})]\cdot \exp[i\varphi_{y}(x_{1},y_{1})]dx_{1}dy_{1}$$
where *z*_1_ is the distance from the observation screen to the diffraction screen. Setting *z*_1_ = *f* and substituting Equations (10)–(12), the change in the electric field can be deduced:(13)$$\vec{E}(x,y)=\frac{\exp(ikz_{1})}{i\lambda z_{1}}\cdot \exp[\frac{i\tau }{2}(x^{2}+y^{2})]\cdot \iint_{\sum }\vec{E}(x_{0},y_{0},z)\cdot \exp[-i\tau (xx_{1}+yy_{1})]+i\frac{\tau^{3}}{3}(m^{3}{x_{1}}^{3}+n^{3}{y_{1}}^{3})]dx_{1}dy_{1}$$
(14)$$\vec{E}(x,y)=\frac{1}{\left|mn\right|}\cdot \frac{\exp(ikz_{1})}{i\lambda z_{1}}\cdot \exp[\frac{i\tau }{2}(x^{2}+y^{2})]\cdot \iint_{\sum }\vec{E}(x_{0},y_{0},z)\cdot Ai(\frac{x}{m})\cdot Ai(\frac{y}{n})dx_{1}dy_{1}$$

As can be seen from Equation (14), the light field of an Airy beam is affected by *m* and *n*. We set *m* = *n* = *a*, where *a* is an any constant. Corresponding to different *a* values under LCP incidence, the xoz and yoz electric field distributions and the intensity distributions at the focus position (the dotted line position in Figure 7a_1_–a_5_ are shown in Figure 7. Figure 7c shows the intensity distribution of the light field along the *x*-axis at the focus. With the increase in the value of *a*, the focal distance will gradually approach the metasurface and the central intensity of the focal position will progressively increase. In other words, the Airy beam becomes more concentrated as the value of *a* increases.

### 3.3. Airy Vortex-like Beams

Next, according to the focus position of the Bessel vortex beam, we correspondingly select the Airy beam with *a* = 16 × 10^−7^. Firstly, considering the case of the general Bessel vortex beam, when LP light is incident, the light fields generated by the incident LCP and RCP will act simultaneously. We slightly adjusted the output intensity of the two beams in order to achieve better fusion under the incidence of LP light. Additionally, we adjust the two light fields to focus on the same plane, and there will be an interaction between the two light fields, resulting in an Airy vortex-like beam, which can be verified by Figure 8. The light field and phase distribution of the *xoy* cross-section at the focal position are shown in Figure 8b,c. It can be found that after the superposition of the light fields, the central spot of the Airy beam is well integrated into the vortex part, forming a series of concentric rings similar to the Airy vortex beam, and its vortex state can be further confirmed in the phase field. The beam combines the singularity properties of vortices and the self-healing properties of Airy beams, which makes it promising for future laser applications.

Finally, under the control of adding HOCP to the Bessel vortex beam, we analyze the polygonal Airy vortex-like beam generated by introducing HOCP. Figure 9 shows the light field schematic diagrams of the *xoz* plane at the focus position, corresponding to the polygonal Airy vortex-like beams generated after the introduction of HOCP from the 3rd order to the 6th order, respectively. The generation of triangular, quadrilateral, pentagonal and hexagonal Airy vortex-like beams is well achieved at the focal position. It can be clearly analyzed in Figure 9 that the Airy beam has been well integrated into the polygonal Bessel beam, and the beam still retains the properties of the Airy beam. This design provides a reference value for the study of special vortex beams in the optical field.

## 4. Conclusions

In summary, we simulated an all-dielectric multifunctional metasurface with high transmittance by FDTD and realized three different functions of generating polygonal Bessel vortex beams, Airy beams and polygonal Airy vortex-like beams. The generation of polygons mainly depends on a new phase structure, namely, HOCP. We confirmed from theoretical calculations and numerical simulations that the metasurface can well shape general vortex beams into arbitrary polygons by controlling the order of the HOCP. Additionally, it is further discovered that the HOCP only changes the light field intensity distribution but not the OAM spectrum. All generated beams have relatively narrow beam widths and extended undiffracted propagation distances. This simple and general approach provides another feasible way to realize the design of multifunctional components. Notably, the functions generated by the metasurface do not interfere with each other, which means the system possesses potential applications in the fields of particle trapping, 3D optical tweezers, parallel laser printing and multiplex fluorescence imaging.

## Figures and Tables

**Figure 1 nanomaterials-13-01094-f001:**
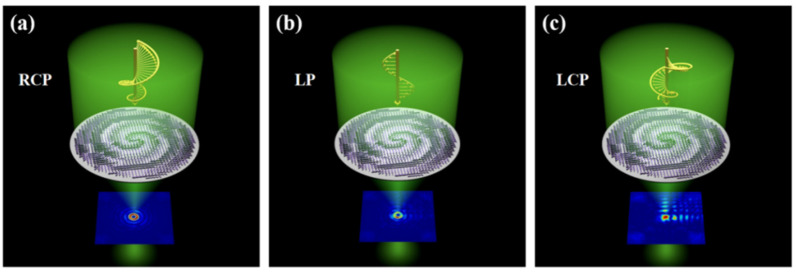
Schematic diagram of multifunctional metasurfaces for realizing non-diffracting beams. (**a**) Generation of polygonal Bessel vortex beam under RCP incidence. (**b**) Generation of Airy vortex-like beam under LP incidence. (**c**) Generation of Airy beam under LCP incidence.

**Figure 2 nanomaterials-13-01094-f002:**
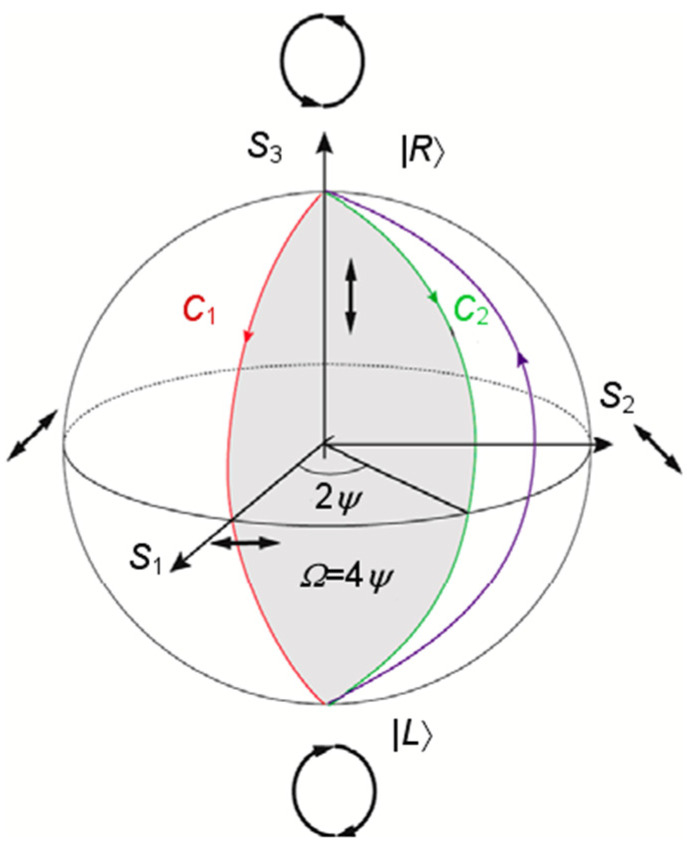
Poincaré sphere model.

**Figure 3 nanomaterials-13-01094-f003:**
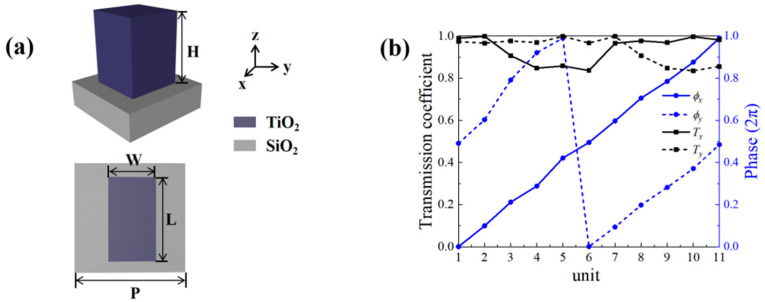
(**a**) Schematic perspective and top view of the TiO_2_ unit structure. (**b**) Corresponding normalized transmission coefficients (black) and phase shifts (blue) for selected 11 units under *x*-LP and *y*-LP incidences.

**Figure 4 nanomaterials-13-01094-f004:**
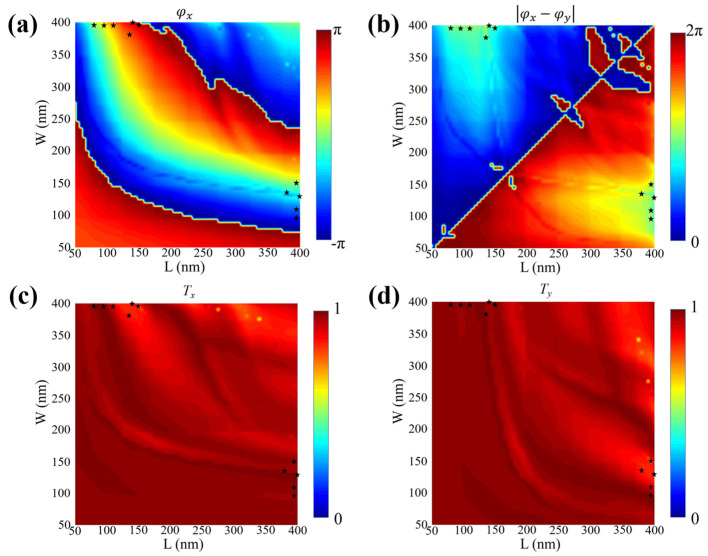
(**a**) Phase shift under *x*-LP incidence. (**b**) Absolute value of the phase offset difference between *x*-LP and *y*-LP. (**c**) Normalized transmission coefficient under *x*-LP incidence. (**d**) Normalized transmission coefficient under *y*-LP incidence. The black pentagrams represent the selected 11 units.

**Figure 5 nanomaterials-13-01094-f005:**
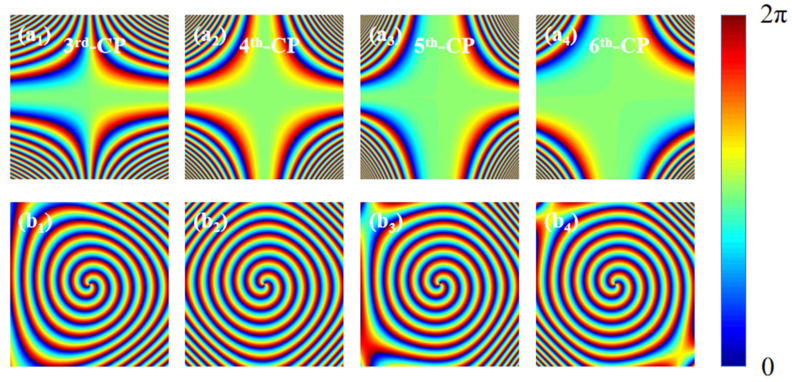
Calculation of phase distribution of Bessel vortex beam modulated by the HOCP. (**a_1_**–**a_4_**) Phase distribution of HOCP. (**b_1_**–**b_4_**) Phase distribution of the Bessel vortex beam after adding HOCP modulation. The parameters *u* in the 3rd-order CP, 4th-order CP, 5th-order CP and 6th-order CP are 3 × 10^15^, 4 × 10^20^, 5 × 10^25^ and 6 × 10^30^, respectively.

**Figure 6 nanomaterials-13-01094-f006:**
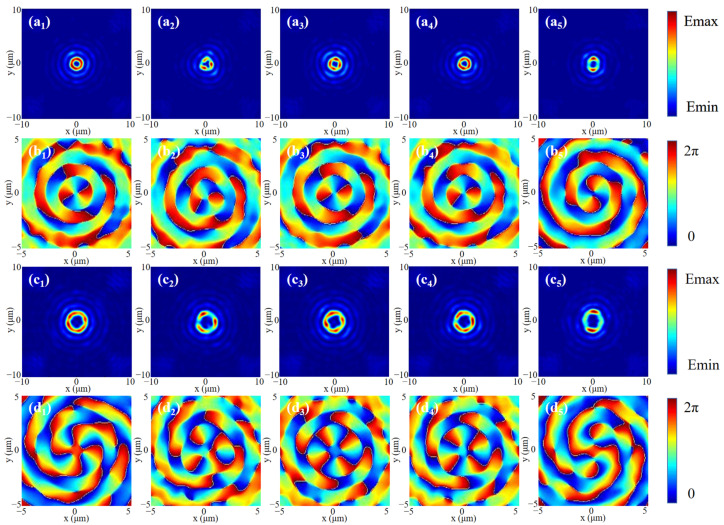
Simulation of the light field and phase distribution of Bessel vortex beams modulated by HOCP under RCP incidence. (**a_1_**–**a_5_**) Optical field distribution of the 2nd-order Bessel vortex beam modulated by HOCP. (**b_1_**–**b_5_**) Phase distribution of the 2nd-order Bessel vortex beam modulated by HOCP. (**c_1_**–**c_5_**) Optical field distribution of the 4th-order Bessel vortex beam modulated by HOCP. (**d_1_**–**d_5_**) Phase distribution of the 4th-order Bessel vortex beam modulated by HOCP. Corresponding to no HOCP, 3rd-HOCP, 4th-HOCP, 5th-HOCP and 6th-HOCP, respectively.

**Figure 7 nanomaterials-13-01094-f007:**
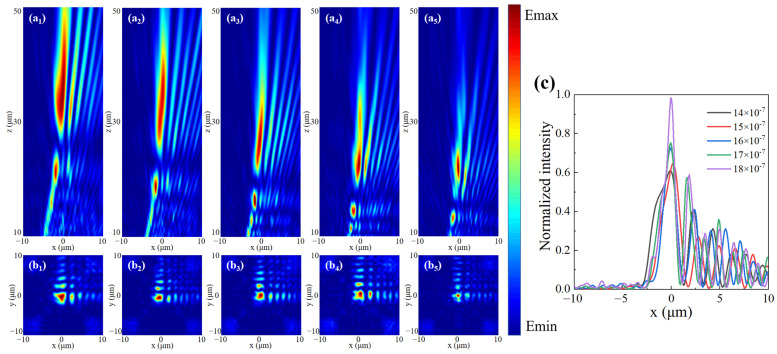
Simulation of Airy beam generation under the LCP incidence for different *a* values. (**a_1_**–**a_5_**) Electric field in the xoz plane. (**b_1_**–**b_5_**) Electric field in the xoy plane at the dashed position. (**c**) Normalized intensity contrast plot at the dotted line.

**Figure 8 nanomaterials-13-01094-f008:**
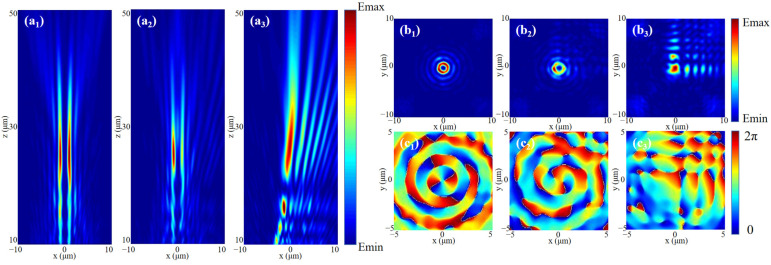
The (**a_1_**–**a_3_**) *xoz* plane light field, (**b_1_**–**b_3_**) *xoy* plane light field and (**c_1_**–**c_3_**) *xoy* plane phase field of the beam generated by RCP incidence, LP incidence and LCP incidence were simulated without HOCP modulation.

**Figure 9 nanomaterials-13-01094-f009:**
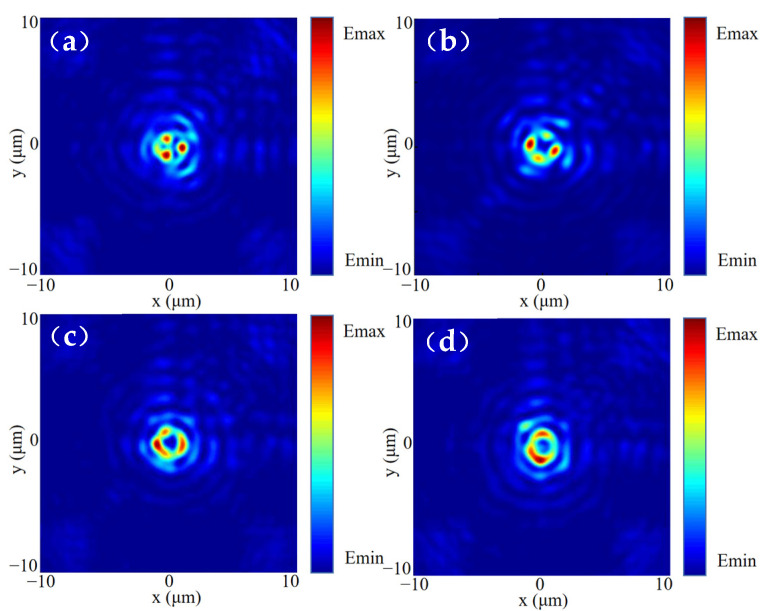
Simulation of polygonal similar Airy vortex beams modulated by adding HOCP, with (**a**) 3rd-HOCP, (**b**) 4th-HOCP, (**c**) 5th-HOCP, and (**d**) 6th-HOCP.

## Data Availability

No new data were created or analyzed in this study. Data sharing is not applicable to this article.

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
