# Peer review of "Numerical Simulation of Integrated Generation and Shaping of Airy and Bessel Vortex Beams Based on All-Dielectric Metasurface"

_nanomaterials, 2023, doi:10.3390/nano13061094_

Round 1

Reviewer 1 Report

The paper is relatively well written and seems scientifically sound. I recommend publishing after minor revisions.

Lines 66-67: "Compared with traditional bulk optical elements, metasurface has the advantages of small size, flexible design adjustment and low manufacturing cost." Whereas I totally agree with the first two advantages, the last one - low manufacturing cost - is superficial, at least with the current technologies, as fabricating of a metasurface is usually done with e-beam lithography, which is pretty expensive and time consuming.

Line 82: Explain "PB phase".

Author Response

The paper is relatively well written and seems scientifically sound. I recommend publishing after minor revisions.

  1. Lines 66-67: "Compared with traditional bulk optical elements, metasurface has the advantages of small size, flexible design adjustment and low manufacturing cost." Whereas I totally agree with the first two advantages, the last one - low manufacturing cost - is superficial, at least with the current technologies, as fabricating of a metasurface is usually done with e-beam lithography, which is pretty expensive and time consuming.

Re: First of all, thank you very much for pointing out the loopholes of this article in time. As for the price of realizing metasurface fabrication, we have not carried out relevant experimental operations due to practical factors, which leads to a lack of consideration in this aspect. In response to this problem, we have revised the original text and changed "Compared with traditional bulk optical elements, metasurface has the advantages of small size, flexible design adjustment and low manufacturing cost." to "Compared with traditional bulk optical elements, metasurface has the advantages of small size and flexible design adjustment."

  1. Line 82: Explain "PB phase"

Re: Thank you very much for your valuable comments. According to your proposal, we have added the description of the PB phase in the 143 to 151 lines, and added the Poincaré sphere model to assist understanding. The additions to the original text are as follows:

The PB phase is based on the adjustment of the phase by rotating the direction of the half-wave plate under circularly polarized light, which can be explained by the Poincaré sphere.[51] As shown in Fig. 2, the north and south levels of the Poincaré sphere respectively represent LCP and RCP, and each point on the equator line corresponds to a different linear polarization state, while the entire spherical surface The other points represent elliptical polarization states for different polarization states. When a point on the sphere rotates an angle Ψ with the center of the sphere as the center, the phase change shown on the sphere is twice the rotation angle, which is the three-dimensional interpretation of the geometric phase.

[51] Berry, M.; The adiabatic phase and Pancharatnam’s phase for polarized light. J. Mod. Optics, 1987, 34, 1401-1407.

Reviewer 2 Report

The authors of the manuscript propose numerical FDTD-based simulations of a multifunctional all-dielectric metasurface comprising anisotropic TiO2 nanopillars on SiO2 substrate. The structure behaves differently for LCP, RCP and LP light. The simulation wavelength is 532 nm (visible light). Potential applications in the fields of optical tweezers and multiplex fluorescence imaging are suggested.

Still, there are some questions that have to be clarified and corrections to be made, as listed below:

  1. As the manuscript presents no experimental verification of the simulation results, I strongly suggest to change the title to “Numerical simulation of integrated generation and shaping of Airy and Bessel vortex beams based on all-dielectric metasurface”
  2. The abbreviations LCP, RCP, LP and PB should be explained where first used. Although relatively common, this will help to avoid possible ambiguity.
  3. Fig. 2 is overloaded with content, subfigures are very small and the axis labels and numbers are hardly visible. I recommend to move Fig. 2 (b)-(e) to a separate figure. For example, it is nearly impossible to discover “the 11 black dots” (row 156) in the figures.
  4. What is the FDTD solver used (row 148)?
  5. What are the values of L and W (row 152) that allow to achieve the multifunctionality claimed by the authors? This is must be clearly explained. If some interesting properties are obtained for the upper limit of the range (400 nm) – that’s where the “black dots” are located, why the simulation is not continued for higher values of L and W?
  6. What physical parameters determine the parameter ‘u’ (row 178)? The values of ‘u’ (rows 201-202) are suspiciously large. How they can be achieved in a real device?
  7. Although the simulation results seem interesting, the lack of any experimental verification seriously diminishes the value of this research.

Author Response

The authors of the manuscript propose numerical FDTD-based simulations of a multifunctional all-dielectric metasurface comprising anisotropic TiO2 nanopillars on SiO2 substrate. The structure behaves differently for LCP, RCP and LP light. The simulation wavelength is 532 nm (visible light). Potential applications in the fields of optical tweezers and multiplex fluorescence imaging are suggested.

Still, there are some questions that have to be clarified and corrections to be made, as listed below:

  1. As the manuscript presents no experimental verification of the simulation results, I strongly suggest to change the title to “Numerical simulation ofintegrated generation and shaping of Airy and Bessel vortex beams based on all-dielectric metasurface”

Re: First of all, thank you very much for this comment. We have revised the title of the original text to:

 “Numerical simulation of integrated generation and shaping of Airy and Bessel vortex beams based on all-dielectric metasurface”.

  1. The abbreviations LCP, RCP, LP and PB should be explained where first used. Although relatively common, this will help to avoid possible ambiguity.

Re: Thank you very much for timely raising this valuable question that we have neglected. According to your comments, we have added the full expression of the abbreviation where it first appeared in this paper. Among them, the modification in the text is:

“The phase design is mainly determined by the combination of the Pancharatnam-Berry (PB) phase adjustment based on Eqs. (5) and the propagation phase adjustment based on Eqs. (7).” (row 141)

  1. 2 is overloaded with content, subfigures are very small and the axis labels and numbers are hardly visible. I recommend to move Fig. 2 (b)-(e) to a separate figure. For example, it is nearly impossible to discover “the 11 black dots” (row 156) in the figures.

Re: Thank you so much for this valuable suggestion. According to your guidance, we have split the original Figure 2 into two figures (Figure 3 and Figure 4), and the split results can be seen from the paper. The modified Figure 4 is shown below, and the problem that “the 11 black dots” were difficult to see has been better solved.

Figure 4. (a) Phase shift under x-LP incidence. (b) Absolute value of the phase offset difference between x-LP and y-LP. (c) Normalized transmission coefficient under x-LP incidence. (d) Normalized transmission coefficient under y-LP incidence.

  1. What is the FDTD solver used (row 148)?

Re: The software we use is FDTD Solutions produced by Canadian Lumerical Solutions, whose full name is the three dimensional finite difference time domain Solutions. FDTD Solutions is a method to numerically calculate different parameters of the electromagnetic field in the time domain based on Maxwell's equations, and use the central difference method to simplify the derivation process and then derive the space-time electromagnetic distribution. More information about the software can be obtained from the link below: https://www.ansys.com/products/photonics/fdtd.

  1. What are the values of L and W (row 152) that allow to achieve the multifunctionality claimed by the authors? This is must be clearly explained. If some interesting properties are obtained for the upper limit of the range (400 nm) – that’s where the “black dots” are located, why the simulation is not continued for higher values of L and W?

Re: Thank you very much for your questions. It may be that the expression of this paper is not clear enough to prevent you from reading well. We first respond to the first small question. The purpose of changing the L and W values is to change the relative refractive index of the material, so as to achieve different refraction effects on the incident beam. Furthermore, according to the difference in response of L and W values to x-polarized and y-polarized light (that is, the mentioned π phase difference) combined with the rotation of the unit structure, the outgoing light in the case of LCP incidence and the output light in the case of RCP incidence are realized. The emitted light is independent of each other. Based on your question, we have added a new description of the paper:

“Figure 3(b) shows the normalized transmittance and phase magnitude of the 11 units selected after the simulation. It can be clearly seen that the selected units satisfy the full coverage of the 2π range well, and has a constant phase response difference of approximately π for x and y polarization. ” (row 175-179)

Secondly, we reply to the second sub-question that "only simulations to 400nm are given in this paper". First of all, through a large number of simulations, we found that when the period is selected to be 400 nm, the effect is the best when the integration and transmission effects are considered at the same time. So we adopted P = 400 nm as the simulation period. Based on the designed unit period of 400 nm, the maximum value of the unit structure can only be simulated at 400 nm. If the simulation range is then expanded, the unit will exceed the boundary of the period. According to your question, we have added a new description of the original text:

“Considering both integration and transmission effects in this design, we set H = 900 nm, P = 400 nm, L and W were varied from 50 nm to 400 nm, respectively.” (row 174)

  1. What physical parameters determine the parameter ‘u’ (row 178)? The values of ‘u’ (rows 201-202) are suspiciously large. How they can be achieved in a real device?

Re: Thank you so much for pointing out an important part of this article that was overlooked when it was written. This u is a parameter used to control the conversion rate. Among them, the magnitude of the u value is determined by the following formula:

It can be seen from the formula that the magnitude of the value of u is mainly used to balance the magnitude difference caused by the relatively small values of x and y in the case of high-order p and q. Because what we are designing is a nanoscale structure, the value of u will be relatively large. Secondly, the specific value of u is determined by a large number of simulations. In our simulation, when the value of u is less than 4*1020, a polygonal beam with a complete closed-loop structure can still be realized at the focal plane. However, as the value of u gradually increases, the beam generated at the focal plane will split due to the high conversion rate, which is similar to the theory related to centrifugal force. Based on your guidance, we have added the following to the paper:

“where λ represents the incident wavelength, β represents the angle between the transmitted wave and z-axis (set to 12.7°), l represents the topological charge, the parameter u controls the beam conversion rate and η is an arbitrary real number.” (row 220-221)

And “It can be seen from Eqs. 9 that the magnitude of the value of u is affected by the order of CP. Since the designed structure is nanoscale, the value of u will be relatively large. ” (row 221-223)

  1. Although the simulation results seem interesting, the lack of any experimental verification seriously diminishes the value of this research.

Re: Thank you for your concern in advance, and we also express our deep regret for your mention of our lack of experiments. At present, due to various practical factors such as funding, we do not support the realization of related experiments for the time being, so the presented articles are only based on simulations. We are also working hard to realize the corresponding results in the experiment as soon as possible.

Reviewer 3 Report

In the paper 'Integrated generation and shaping of Airy and Bessel vortex  beams based on all-dielectric metasurface' the authors present their recent results about vortex shaping by dielectric metasurfaces. The paper is well organized and the results are clearly presented. I have only few minor comments which I list here:

1. I think the authors should better comment about practical realization of their design? What are the main difficulties? Is this design easily achievable?

2. The same holds for the experimental part. I think the authors shoud better describe how it should be possible to verify their theoretical results

Author Response

In the paper 'Integrated generation and shaping of Airy and Bessel vortex  beams based on all-dielectric metasurface' the authors present their recent results about vortex shaping by dielectric metasurfaces. The paper is well organized and the results are clearly presented. I have only few minor comments which I list here:

  1. I think the authors should better comment about practical realization of their design? What are the main difficulties? Is this design easily achievable?

Re: First of all, thank you very much for your recognition of our article. In response to your first question, we have included in the original article our relevant comments on the designed metasurface:

“Compared with other multifunctional metasurfaces, we only use the difference in polarization sensitivity to realize the generation of three kinds of non-diffracting beams, and add beam shaping function on the basis of traditional vortex, which has certain reference significance for the development of multifunctional metasurfaces. Further, since the modulated beam has a stable distribution at the far-field focus, it is of great significance to particle trapping, 3D optical tweezers, parallel laser printing and multiplex fluorescence imaging.” (row 96-100)

Next, we will answer the second and third questions. During our attempts, the biggest problem we encountered was that the superposition effect of the beams could not achieve the expected results when linearly polarized light was incident. Therefore, we constantly adjust the intensity of the outgoing beams generated under the incident conditions of left-handed circularly polarized light and right-handed circularly polarized light to control the situation of the beam generated under the incident linearly polarized light. Overall, this design is not easy to implement. In response to this question, we add the following description to the original text:

“We slightly adjusted the output intensity of the two beams in order to achieve better fusion under the incidence of LP light. ” (row 292-293)

  1. The same holds for the experimental part. I think the authors shoud better describe how it should be possible to verify their theoretical results

Re: Thank you very much for your valuable comments. We first introduce that all the simulation parts use FDTD solutions software to support the theoretical results. FDTD Solutions is a method to numerically calculate different parameters of the electromagnetic field in the time domain based on Maxwell's equations, and use the central difference method to simplify the derivation process and then derive the space-time electromagnetic distribution. Therefore, after deriving the results from the numerical theory, we simulated all the theoretical results through scripts in FDTD, and presented the final results in the article. In response to your question, we have added a certain description to the original text:

“We use scripts to import the above theoretical content into FDTD, and obtain corresponding results under 3D simulation to support the above theoretical derivation.” (row 168-169)

Round 2

Reviewer 2 Report

Reading the revised manuscript, I can see that the authors did their best to take into account reviewers’ comments and recommendations, which has improved the quality of the manuscript. I am also satisfied by the authors responses to my questions. Hence, my opinion is that the revised manuscript can be published as it is.